# Adjunct Therapy with T Regulatory Cells Decreases Inflammation and Preserves the Anti-Tumor Activity of CAR T Cells

**DOI:** 10.3390/cells12141880

**Published:** 2023-07-18

**Authors:** Ke Zeng, Meixian Huang, Mi-Ae Lyu, Joseph D. Khoury, Sairah Ahmed, Krina K. Patel, Boro Dropulić, Jane Reese-Koc, Paolo F. Caimi, Tara Sadeghi, Marcos de Lima, Christopher R. Flowers, Simrit Parmar

**Affiliations:** 1Department of Lymphoma & Myeloma, The University of Texas MD Anderson Cancer Center, Houston, TX 77030, USA; kzeng@mdanderson.org (K.Z.); mhuang8@mdanderson.org (M.H.); mlyu@mdanderson.org (M.-A.L.); sahmed3@mdanderson.org (S.A.); kpatel1@mdanderson.org (K.K.P.); crflowers@mdanderson.org (C.R.F.); 2Department of Pathology and Microbiology, University of Nebraska Medical Center, Omaha, NE 68198, USA; jkhoury@unmc.edu; 3Caring Cross, Gaithersburg, MD 20878, USA; boro.dropulic@caringcross.org; 4Department of Cellular Therapy, School of Medicine, Case Western Reserve University, Cleveland, OH 44106, USA; jane.reese@case.edu; 5Department of Hematology and Medical Oncology, Cleveland Clinic, Cleveland, OH 44195, USA; paolo.caimi@case.edu; 6Cellenkos Inc., Houston, TX 77005, USA; tara.sadeghi@cellenkosinc.com; 7Division of Hematology, The Ohio State University College of Medicine, Columbus, OH 43210, USA; marcos.delima@osumc.edu

**Keywords:** regulatory T cells, CAR T cells, CRS, lymphoma, allogeneic, umbilical cord blood

## Abstract

With greater accessibility and an increased number of patients being treated with CAR T cell therapy, real-world toxicity continues to remain a significant challenge to its widespread adoption. We have previously shown that allogeneic umbilical cord blood-derived (UCB) regulatory T cells (Tregs) can resolve inflammation and treat acute and immune-mediated lung injuries. Allogeneic, cryopreserved UCB Tregs have shown a clinical benefit in patients suffering from COVID-19 acute respiratory distress syndrome. The unique properties of UCB Treg cells include a lack of plasticity under inflammatory micro-environments, no requirement for HLA matching, a long shelf life of cryopreserved cells, and immediate product availability, which makes them attractive for treating acute inflammatory syndromes. Therefore, we hypothesized that adjunct therapy with UCB Tregs may resolve the undesirable inflammation responsible for CAR T cell therapy-associated toxicity. In in vitro analysis, no interference from the addition of UCB Tregs was observed on CD19 CAR T cells’ ability to kill CD19 Raji cells at different CAR T: Raji cell ratios of 8:1 (80.4% vs. 81.5%); 4:1 (62.0% vs. 66.2%); 2:1 (50.1% vs. 54.7%); and 1:1 (35.4% vs. 44.1%). In the xenogeneic B-cell lymphoma model, multiple injections of UCB Tregs were administered 3 days after CD19 CAR T cell injection, and no detrimental effect of add-on Tregs was noted on the circulating CD8^+^ T effector cells. The distribution of CAR T cells in multiple organs remained unaffected by the addition of the UCB Tregs. Specifically, no difference in the overall tumor burden was detected between the UCB Treg + CAR T vs. CAR T alone recipients. No tumor was detected in the liver or bone marrow in CAR T cells + UCB Tregs recipients, with a notable corresponding decrease in multiple circulating inflammatory cytokines when compared to CART alone recipients. Here we show the proof of concept for adjunct therapy with UCB Tregs to mitigate the hyper-inflammatory state induced by CAR T cells without any interference in their on-target anti-tumor activity. Administration of UCB Tregs after CAR T cells allows sufficient time for their synapse formation with tumor cells and exerts cytotoxicity, such that the UCB Tregs are diverted to interact with the antigen-presenting cells at the site of inflammation. Such a differential distribution of cells would allow for a two-pronged strategy of a UCB Treg “cooling blanket” effect and lay the groundwork for clinical study.

## 1. Introduction

To control tumor growth, chimeric antigen receptor (CAR) T cells bind target cell surface antigens through a single-chain variable fragment (scFv) recognition domain, forming a non-classical immune synapse required for their effector function, where they mediate their anti-tumoral effects through the perforin and granzyme axis, the Fas and Fas ligand axis, as well as the release of cytokines to sensitize the tumor stroma [1]. The potent systemic immune activation responsible for the success of CAR T cells also drives the life-threatening toxicity of cytokine release syndrome (CRS) [2], neurotoxicity, and impaired hematopoietic recovery that impose significant morbidity and mortality [3]. Continued efforts in managing such side effects, including steroids, Tocilizumab, and anti-IL1a antagonists, have led to substantial improvement in patients’ clinical outcomes, such that multiple CAR T cell therapies have been FDA-approved for the treatment of life-threatening hematologic malignancies. However, with greater accessibility and an increased number of patients being treated with CAR T cell therapy, real-world toxicity continues to remain a significant challenge to the widespread adoption of such a lifesaving therapy.

T regulatory cells (Tregs), an Immunoregulatory subset of T cells, play an important role in maintaining self-tolerance, dampening the immune response to foreign antigens, resolving excessive inflammation, and maintaining immunological homeostasis [4,5]. However, these circuits can be co-opted by tumors in tissue-specific and anatomic compartment-restricted manners, where Tregs can be recruited to the tumor site, which in turn can promote tumor growth by hindering the effector immune response [6]. On the other hand, some studies show that the presence of Tregs can be correlated with a positive prognosis, especially in inflammatory malignancies such as head-and-neck cancer [7]. Therefore, Treg cells are emerging as a central controller of the thin balance between autoimmunity and cancer.

We have previously shown that allogeneic umbilical cord blood-derived (UCB) Treg cells can resolve uncontrolled inflammation and treat acute and immune-mediated lung injury in a xenogeneic model [8]. Such findings have been translated into a clinical setting, where, in patients suffering from COVID-19 acute respiratory distress syndrome (ARDS), UCB Treg cell infusions lead to the control of systemic inflammation correlated with clinical improvement [9]. In another randomized placebo control trial, allogeneic, non-HLA-matched, cryopreserved UCB Tregs administered intravenously in a fixed dose of 100 million cells on days 0, 3, and 7 to intubated and mechanically ventilated patients with COVID-associated moderate to severe acute respiratory distress syndrome led to improvement in overall survival when adjusted for covariates including age, gender, race, vasopressors, oxygenation, and duration of intubation [10]. Furthermore, UCB Treg cells have been shown to prevent graft vs. host disease (GVHD) in preclinical [11] and clinical settings [12,13,14]. The unique properties of UCB Treg cells, including: (i) lack of plasticity when exposed to inflammatory micro-environments [8]; (ii) no requirement for HLA matching with the recipients [9,10]; (iii) long shelf life of the cryopreserved cells; and (iv) immediate product availability for on-demand treatment, make them attractive for treating acute inflammatory syndromes. Therefore, we hypothesized that adjunct therapy with UCB-derived Treg cells may resolve the undesirable inflammation responsible for CAR T cell therapy-associated toxicity. Here we provide proof-of-concept studies supporting the non-interference of allogeneic UCB Treg cells on the on-target anti-tumor action of CD19 CAR T cell therapy while dampening systemic inflammation.

## 2. Materials and Methods

### 2.1. Cell Line, UCB Treg Cell Generation, and CD19 CAR-T Cells

UCB units were obtained under The University of Texas MD Anderson Cancer Center’s (MDACC) Institutional Review Board-approved protocols (IRB2020-1134). CD25^+^ Treg cells were isolated and expanded from the UCB units as described previously [8,11]. After 14 days of culture, ex vivo expanded UCB-derived Treg cells were harvested and frozen. Additionally, cryopreserved UCB Treg cells were provided by Cellenkos^®^ Inc. (Houston, TX, USA). The UCB Tregs provided by the two different sources were comparable in phenotype and suppressor function, as described previously [8]. The clinical-grade CD19 CAR T cells were obtained from the Cellular Therapy Laboratory, Seidman Cancer Center, Case Western Reserve University, and were manufactured using ClinicMACS Prodigy as per the manufacturer’s instructions (Miltenyi Biotech, Gladbach, Germany). The RFP-fLuc dual-reporters-labeled Raji cell line was a gift from Dr. Sattva Neelapu’s lab at MDACC. No HLA matching was performed between the UCB Tregs and CD19 CAR T cells or Raji B cells [8].

### 2.2. Flow Analysis of CAR-T Cells and Ex Vivo Expanded UCB-Derived Treg Phenotype

At the end of the 14 day culture, phenotypic analysis of the ex-vivo expanded UCB Treg cells was performed by analysis of surface or intracellular markers, using the following monoclonal antibodies: anti-hCD4/BUV737 (clone SK3, Becton Dickinson, Franklin Lakes, NJ, USA), anti-hCD8/FITC (clone SK1, Becton Dickinson, Franklin Lakes, NJ, USA), anti-hCD25/PE (clone 2A3, Becton Dickinson, Franklin Lakes, NJ, USA), anti-hCD45/APC (clone 2D1, Becton Dickinson, Franklin Lakes, NJ, USA), and anti-mCD45/BV605 (clone 30-F11, Becton Dickinson, Franklin Lakes, NJ, USA) for surface staining; anti-FoxP3/PE-Cy7 (clone 236A/E7, Thermo Fischer Scientific, Waltham, MA, USA), anti-Helios/APC (clone 22F6, Thermo Fischer Scientific, Waltham, MA, USA) for intracellular staining; FOXP3 Fix/Perm buffer set (BioLegend, SanDiego, CA, USA) was used for intracellular staining according to manufacturer’s instructions. LIVE/DEAD™ Fixable Dead Cell Stain (Thermo Fischer Scientific, Waltham, MA, USA) was used as a viability dye for live cell gating. FITC-Labeled Human CD19 (20-291) and Fc Tag (AcroBiosystems, Newark, DE, USA) were used to analyze the chimeric antigen receptor expression of CAR-T cells. Stained cells were acquired on a BD LSR Fortessa X-20 flow cytometer (Becton Dickinson, Franklin Lakes, NJ, USA) and analyzed using FlowJo^TM^ software ver.10 (Ashland, OR, USA).

### 2.3. Treg Cell Suppression Assay

To examine the functional ability of ex-vivo expanded UCB Treg cells, CD4^+^CD25^−^ conventional T cells (Tcons) were stained with CellTrace^TM^ Violet (CTV) (Thermo Fisher Scientific, Waltham, MA, USA) as described previously [8]. Proliferation of CTV-labeled Tcons was assessed by the LSR Fortessa Cell Analyzer after 96 h of culture.

### 2.4. In Vitro CAR T Cell Cytotoxicity Assay

Cytotoxic-specific killing of RFP-Raji cells by CD19 CAR T cells was analyzed by a flow cytometry-based assay as previously described [15]. Briefly, RFP-labeled Raji cells were counted to guarantee >90% viability prior to assay and plated at 1 × 10^4^ cells/well. CD19 CAR T cells were then added at the indicated E:T ratios. After 24 h of co-culture, a fixed volume of mixed cell culture was collected from a 96-well plate to count live single RFP-positive Raji cells by flow cytometry using RFP (mCherry) gating. The lysed Raji cell number (starting Raji cell number—live Raji cell number after 24 h of co-culture) and lysis rate can be calculated.

### 2.5. In Vivo Xenogeneic Lymphoma Mouse Model

Animal procedures were performed according to an approved protocol by the MDACC Institutional Animal Care and Use Committee (IACUC 00001051-RN02). The xenogeneic lymphoma model was generated using female Rag2-γc-mice supplied by The Jackson Laboratory at 5 weeks of age. Female Rag2-γc-mice were transplanted with 0.3 × 10^6^ RFP-fLuc-Raji cells by tail vein injection. After mice displayed engraftment of tumor mass as per in vivo imaging, a one-time tail vein injection of 4.5 × 10^6^ CD19 CAR T cells was administered on day 4, followed by three tail vein injections of 1 × 10^7^ UCB Treg cells administered on days +7, +9, and +11. Mice were imaged using non-invasive bioluminescence for tumor burden as well as assessed weekly for tumor progression (Section 2.7). The proliferation level of CAR T cells in the mouse peripheral blood (PB) was analyzed by flow cytometry. Mice were monitored twice per week for weight loss and survival. The dose of Raji cells and CD19 CAR T cells was based on published literature, followed by additional titration in our laboratory [16]. The dose of UCB Treg cells was based on our laboratory experience [8].

### 2.6. Preparation and Phenotype Analysis of PBMCs, Hepatocytes, Splenocytes, and Bone Marrow Cells

The PB of lymphoma xenografts was collected into ethylenediaminetetraacetic acid (EDTA)-coated tubes. A 100-microliter sample of heparinized blood was incubated with 1x RBC lysis buffer (Miltenyi Biotec, Germany), washed with phosphate-buffered saline (PBS), and centrifuged to collect the PB mononuclear cells (PBMCs). The liver, spleen, and bone marrow (BM) of lymphoma xenografts were harvested, homogenized, and filtered using a nylon mesh to obtain single-cell suspensions. After red blood cell (RBC) lysis, hepatocytes, splenocytes, and BM cells were collected by centrifugation. The phenotypic characterization of cells was performed by the analysis of surface markers. Ex vivo expanded UCB-Tregs, CAR T cells, or Raji cells isolated from PB, liver, spleen, and BM of lymphoma xenografts were assessed using the following Abs: anti-hCD4/BUV737 (clone SK3, Becton Dickinson, Franklin Lakes, NJ, USA), anti-hCD8/FITC (clone SK1, Becton Dickinson, Franklin Lakes, NJ, USA), anti-hCD25/PE (clone 2A3, Becton Dickinson, Franklin Lakes, NJ, USA), anti-hCD45/APC (clone 2D1, Becton Dickinson, Franklin Lakes, NJ, USA), and anti-mCD45/BV605 (clone 30-F11, Becton Dickinson, Franklin Lakes, NJ, USA). FITC-Labeled Human CD19 (20-291) and Fc Tag (AcroBiosystems, Newark, DE, USA) were used to analyze the chimeric antigen receptor expression of CAR-T cells. Data acquired by the Fortessa X-20 Cell Analyzer (BD Biosciences) were analyzed with BD FACSDiva 8.0.1 software and FlowJo^TM^ software. A complete blood count was also performed on the mouse PB.

### 2.7. Non-Invasive Bioluminescence Imaging

Raji tumor burden in mice was evaluated by Non-invasive bioluminescence imaging (BLI) with the IVIS Lumina X5 Imaging System (PerkinElmer, Waltham, MA, USA). Prior to imaging, RFP-fLuc-Raji-engrafted mice were injected intraperitoneally with D-luciferin solution (150 mg/kg; Becton Dickinson, Franklin Lakes, NJ, USA) and anesthetized with isoflurane using a rodent anesthesia system (PerkinElmer, Waltham, MA, USA). After 10 min of D-luciferin injection, mice were imaged, and the bioluminescent signals were acquired and quantified using Living Image 4.7.3 (PerkinElmer, Waltham, MA, USA).

### 2.8. Cytokine Analysis

Plasma samples were collected weekly from the EDTA-treated mice PBs. Levels of human EGF, Eotaxin-1, FGF-2, Flt-3L, Fractalkine, G-CSF, GM-CSF, GRO-α, IFN-α2, IFN-γ, IL-1α, IL-1β, IL-1rα, IL-2, IL-3, IL-4, IL-5, IL-6, IL-7, IL-8, IL-9, IL-10, IL-12 (p40), IL-12 (p70), IL-13, IL-15, IL-17A, IL-18, IP-10, MCP-1, MCP-3, MDC, MIP-1α, MIP-1β, PDGF-AA, PDGF-AB/BB, RANTES, sCD40L, TGF-α, TNF-α, TNF-β, and VEGF-A in the mouse plasma were measured using the Human Cytokine 42-plex Discovery Assay Kit (Eve Technologies, Calgary, AB, Canada).

### 2.9. Histopathology and Immunohistochemistry

Liver and BM tissue were harvested from the euthanized mice, fixed with 10% buffered formalin, and embedded in paraffin for processing into 5 μm tissue sections in the Research Histology Core Laboratory at MDACC. To assess pathology and cellular infiltrates in the lymphoid tissues, de-paraffinized and rehydrated tissue sections were stained with Hematoxylin and Eosin and evaluated by a pathologist. For immunohistochemistry, deparaffinized and rehydrated tissue sections were subjected to heat-mediated antigen retrieval with sodium citrate buffer (pH 6), permeabilization, and blocking prior to staining with primary antibodies to human CD3, CD4, CD8, or CD19. Appropriate horseradish peroxidase-conjugated secondary antibodies were used to determine the subset of human cells in the tissues of mice (PDX model). The stained tissue slides were analyzed under Aperio ImageScope (Leica Biosystem, Buffalo Grove, IL, USA).

### 2.10. Statistical Analysis

All statistical analyses were performed with GraphPad Prism 9 software (version 9.5.0) (San Diego, CA, USA). Data are presented as mean ± SEM. *p* values were calculated using a two-tailed *t*-test with 95% confidence interval, one-way ANOVA, or two-way ANOVA for evaluation of statistical significance compared with the untreated controls. *p* < 0.05 was considered statistically significant.

## 3. Results

### 3.1. Co-Culture with UCB Tregs Does Not Impact CAR T Cell Cytotoxicity

Freshly harvested UCB Tregs are routinely characterized after 14 days of ex-vivo culture with CD3/28 Dynabeads^TM^ and IL-2. Flow analysis demonstrates that >80% of the expanded UCB Treg cells express a consistent phenotype of CD4^+^CD25^+^CD127^lo^FOXP3^hi^Helios^hi^ (Figure 1A) and were able to suppress proliferating Tcon (CD4^+^CD25^−^) cells (Figure 1B). Phenotypic analysis of clinical-grade CD19 CAR T cells gated on lymphocytes with exclusion of Treg cells revealed CAR expression in 57%, single CD4^+^ in 80.9%, and single CD8^+^ in 12.1% (Figure 1C). RFP-labeled CD19^+^ Raji cells exhibited high fluorescence of 97.8% (Figure 1D). CD19 CAR T cells exerted significantly higher lysis of Raji cells when compared to the lysis exerted by Tcons at different E:T cell ratios (Figure 1E). In order to understand the impact of the addition of UCB Tregs on the ability of CD19 CAR T cells to exert cell lysis, UCB Tregs were added in a 1:1 ratio to the CAR T cells in the above cytotoxicity assay, and no impact of the addition of UCB Tregs was observed in the ability of the CD19-CAR T cells to exert cytotoxicity on CD19 Raji cells at different ratios (Figure 1F).

### 3.2. UCB Treg Cells Do Not Interfere in CAR T Cell Persistence In Vivo

In order to understand the in vivo kinetics of the interaction of UCB Treg cells with CAR T cells, we generated a xenogeneic lymphoma model where the disease was established by the tail vein (t.v.) injection of 0.3 × 10^6^ RFP-fLuc dual-reporters-labeled Raji cells on Day 0. Four experimental arms were established: Arm 1: Control arm with Raji cells alone; Arm 2: Raji cells + UCB Tregs given on days +7, +9, and +11; Arm 3: Raji cells followed 3 days later by t.v. injection of 4.5 × 10^6^ CD19 CAR T cells; and Arm 4: Raji cells followed 3 days later by t.v. injection of 4.5 × 10^6^ CD19 CAR T cells where multiple t.v. injections of 1 × 10^7^ UCB Treg cells were administered on days 7, 9, and 11 (Figure 2A). The dosing and timing of UCB Tregs were chosen based on previously published data, where a fixed dose of 1 × 10^7^ UCB Treg cells injected every 2 days was able to treat acute lung injury, as was weekly dosing for immune-mediated lung injury [8]. Furthermore, in the clinical setting, two to three doses of UCB Treg cells administered at a fixed dose of 100 million cells at a 48-h window were well tolerated in two patients suffering from multi-organ failure of COVID ARDS and were associated with an early efficacy signal [9,10]. The benefit of such a fixed dose strategy is further supported by early survival data in a randomized placebo-controlled trial of CK0802 (a cryopreserved, allogeneic, HLA-unmatched, multi-dose, UCB Treg cell product) [10]. A single t.v. injection of 0.3 × 10^6^ Raji cells led to the development of disease in NSG mice, as demonstrated on non-invasive bioluminescence imaging, both in Arm 1 and Arm 2 (Figure 2B). Prior to their injection, UCB Treg cells were labeled with the intracellular CTV dye, which emits a fluorescent signal detectable by flow cytometry for at least 7 days. As shown in Figure 2C (left panel), 99.3% of the circulating non-tumor cell population in Arm 2 is comprised of CTV-labeled Treg cells with a phenotype of CD4^+^CD25^+^. As shown in Figure 2D, on day +17, the injected UCB Treg cells were distributed primarily to peripheral blood and spleen (60%) followed by liver and bone marrow tissue (20%).

On day 14 after CAR T cell injection (day +18 after Raji cell injection), mouse PB was analyzed for circulating T cell populations, including CD4 and CD8 cells in Arm 3 (C) and Arm 4 (C+T). As shown in Figure 3A,B (representative graphs), the circulating CD8^+^ effector T-cells were comparable in the two arms (Figure 3B). On Day +23 (and D+19 since CAR T cell injection), no differences were observed in the distribution of the CD8^+^ effector population (Figure 3C) or the CAR expression (Figure 3D) in liver, spleen, and BM samples of Arm 3 (C) and Arm 4 (C+T) mice. Similar to our previous report [8], the circulating UCB Treg cells remained elevated in the PB circulation until day +14 (since the first UCB Treg cell injection) and dropped to control levels around day +18 in Arm 4 (C+T) mice (Figure 3E). However, the presence of UCB Treg cells in the peripheral circulation did not negatively impact the circulating CAR T cells as measured on days +14 and +21 (Figure 3F).

### 3.3. UCB Treg Cells Do Not Impact the Target Efficacy of CD19 CAR T Cells

To further understand the impact of the multiple UCB Treg cell injections on CAR T cell tumoricidal efficacy in vivo, we performed planned euthanasia and compared the four arms: Arm 1, Raji only; Arm 2, Raji + UCB Treg cells; Arm 3, Raji + CAR T; Arm 4, Raji + CAR T + UCB Tregs. Specifically, liver, spleen, and bone marrow tissues were analyzed. As shown in Figure 4A–C, the liver architecture was disrupted and invaded by the tumor cells in Arm 1 and Arm 2. Immunohistochemical staining showed the detection of CD19-positive tumor cells in both arms (Figure 4B). No evidence of a tumor was detected in Arm 3 or Arm 4, with complete preservation of the tissue architecture. Specifically, no differences in the CD3+ T cell infiltrate were observed in either arm. Quantification of CD19-positive cells showed no differences in Arm 1 vs. Arm 2; however, a significant decrease was observed in Arm 3 vs. Arm 1 and Arm 4 vs. Arm 1. No differences were observed between Arm 3 and Arm 4. Additionally, quantification analysis of the H-score for human CD19^+^ was also performed, where a high score was observed for both Arm 1 and Arm 2, with a significant drop for both Arm 3 and Arm 4. No differences were observed between Arm 3 and Arm 4. A similar impact was also observed in another set of control and treated mice (Appendix A).

As shown in Figure 4E, although tissue disruption was seen in the spleen of Arm 1: Raji only control mice and Arm 2: Raji + UCB Treg cells, tumor invasion was not detected by the CD19 IHC stain (Figure 4F). CD3 T cell staining showed some presence in Arm 2: Raji + UCB Treg only; Arm 3: Raji + CART recipients; and Arm 4: Raji + CAR T + UCB Treg cell recipients (Figure 4G). Quantification of CD3^+^ cells showed a decrease in Arm 4: Raji + CART + Treg cell recipients when compared to Arm 3: Raji + CART alone or Arm 2: Raji + UCB Tregs arm (Figure 4H). CD3 H-score analysis also confirmed the decrease in CD3+ cells in Arm 4: Raji + CART + Treg arm when compared to Arm 3: Raji + CAR T cells alone (Figure 4H). Similar findings were also observed in another set of control and treated mice (Appendix A). Lastly, as shown in Figure 4I, massive tumor infiltration of bone marrow was detected in Arm 1: Raji control and Arm 2: Raji + UCB Treg cell recipients, whereas no tumor was detected in Arm 3: Raji + CAR T and Arm 4: Raji + CAR T + UCB Treg cell recipients. A dense lymphocytic infiltration was evident in Arm 3: Raji + CAR T recipient, whereas the bone marrow architecture was well preserved in Arm 4: Raji + CAR T + UCB Treg cell recipient. CD19 IHC stain was dense in both Arm 1: Raji control and Arm 2: Raji + UCB Treg recipients and scattered in Arm 3: Raji + CART and Arm 4: Raji + CART + Treg recipients (Figure 4J). CD3 infiltrate was absent in Arm 1: Raji control; scattered in Arm 2: Raji + UCB Treg recipients; increased in Arm 3: Raji + CAR T cell recipients; and dense in Arm 4: Raji + CAR T + Treg cell recipients (Figure 4K). CD19 quantification and H-score analysis showed a significant decrease in Arm 3: Raji + CAR T cell and Arm 4: Raji + CAR T + Treg cell recipients when compared to Arm 1: Raji only control and Arm 2: Raji + Treg cell recipients (Figure 4L), whereas no differences were seen between Arm 3: Raji + CAR T cell and Arm 4: Raji + CAR T + Treg cell recipients. CD3 quantification showed a statistically significant increase in the Arm 4: Raji + CART + Treg cell recipient when compared to the Arm 3: Raji + CAR T cell-only recipient (Figure 4M). Scattered CD3 IHC staining was observed in Arm 3: Raji + CAR T recipient, with a slightly lesser burden in Arm 4: Raji + CAR T + UCB Treg cell recipient (Appendix A).

Non-invasive bioluminescence showed clear evidence of disease progression in Arm 1: Raji controls and Arm 2: Raji + UCB Treg cell recipients, where these mice had to be euthanized by day 19. For real-time comparison, one mouse from Arm 3: Raji + CAR T and Arm 4: Raji + CAR T + UCB Treg cell arm were also sacrificed. (Figure 5A). Non-specific dye uptake was detectable by imaging studies on day 17 since no histologic evidence of disease was observed in Arm 3: Raji + CAR T and Arm 4: Raji + CAR T + UCB Treg cell recipients (Figure 4). Subsequently, by day 52, tumor progression was evident in both Arm 3: Raji + CAR T and Arm 4: Raji + CAR T + UCB Treg cell recipients (Appendix A). No differences in the bioluminescence quantification were observed in Arm 3: Raji + CART vs. Arm 4: Raji + CART + Tregs up to Day 52 (Figure 5B). Both Arm 3: Raji + CAR T and Arm 4: Raji + CAR T + UCB Treg cell recipients maintained their baseline weight over time (Figure 5C,D).

### 3.4. UCB Treg Cells Decrease CAR T Cell-Induced Off-Target Inflammatory Response

Since CAR T cells are notorious for causing off-target morbidity primarily driven by the hyper-inflammatory response as well as the cytokine release syndrome, we systematically examined the impact of the “add-on therapy” of UCB Treg cells post-CAR T cell injection and analyzed PB samples for circulating inflammatory cytokines on day 14 of planned euthanasia. As shown in Figure 6A, Arm 3: Raji + CAR T cell recipients (+C+T) showed increased levels of multiple inflammatory cytokines compared to Arm 1: Raji only control arm (R). These inflammatory cytokines were significantly decreased in Arm 3: Raji + CAR T + UCB Treg cell recipients (+C+T): (i) Eotaxin; (ii) GM-CSF; (iii) IFN-α; (iv) IFN-γ; (v) IL-13; (vi) MCP-1; (vii) MCP-3; and (viii) soluble CD40 ligand (sCD40L). Specifically, when compared to Arm 3: Raji + CAR T cell recipients (+C), Arm 4: Raji + CAR T + UCB Treg cells (+C+T) clearly decreased the burden of several inflammatory cytokines (Figure 6B).

## 4. Discussion

Here we present a proof of concept for adjunct therapy with UCB Treg cells to mitigate the hyper-inflammatory state induced by CAR T cells in a xenogeneic lymphoma model without interfering with the CAR T cells’ on-target anti-tumor activity. Our preclinical data lays the groundwork for examining this hypothesis in a clinical setting.

While tumor-resident Treg cells are believed to be responsible for tumor evasion and treatment resistance, the use of UCB Treg-based therapy to treat CAR T cell-induced CRS might seem counterintuitive [17]. However, we did not observe any deleterious effect of adding UCB Treg cells on CAR T cells’ ability to exert a cytotoxic effect on Raji B-cell lymphoma cells in vitro or in vivo. Possible explanations for their non-interference could be: (i) allogeneic, healthy UCB Treg cells may preferentially migrate towards and primarily engage with antigen-presenting cells (APCs) at the site of inflammation [18], and (ii) the 3-day interval between CAR T cell injection and UCB Treg cell administration might allow sufficient time for the CAR T cells to form synapses with the tumor cells [1], which remain undisturbed by the UCB Treg cells. As such, CAR T cell-induced CRS is typically observed approximately 2–3 days after their injection [19], and therefore, the timing of UCB Treg cell injection might facilitate their selective distribution towards APCs, exerting a “cooling blanket” effect at the site of inflammation.

Consistent with published data [16], in our xenogeneic lymphoma model, the CD19 CAR T cells accumulated at the sites of tumor, including peripheral blood, liver, spleen, or bone marrow. The addition of the UCB Treg cells did not impact CAR expression or CAR T cell persistence in these organs. Tumor cell invasion of the liver was resolved in both CAR T and CAR T + UCB Treg cell recipients. Notably, there were no differences in the overall tumor burden as measured by non-invasive bioluminescence and no difference in overall survival between the CAR T vs. CAR T + UCB Treg cell recipients, providing further support for the consideration of adoptive therapy with UCB Treg cells as a strategy to mitigate the unwanted inflammatory damage caused by CAR T cells.

Although our CD19 CAR T cells showed a dominance of CD4^+^ cells in vitro, the killing mechanisms were still correlated with the CD8^+^ cytotoxic cells in vivo [20], where CAR T cells have been shown to undergo dynamic changes during the process of killing tumors in vivo, such that CAR molecule consumption may lead to a decrease in CAR expression, and CAR T cells may further differentiate towards effector-like CD8^+^ T cells. We specifically examined and found no impact of UCB Treg cells on the distribution of CD8^+^ effector/cytotoxic T cells in CAR T cell recipients. In our xenogeneic lymphoma model, the highest tumor burden was detected in the bone marrow. We observed an increase in bone marrow CD3^+^ cells in CART + UCB Treg recipients, which could be a result of the CAR T cell-mediated clearance of tumor cells [21]. Although the synergy of UCB Tregs with CD19 CAR T cell-mediated tumor kill could not be established in our model, a recent study showed a significant early increase in the levels of CD4^+^CD25^+^FoxP3^+^ Tregs in leukemic hu-mice treated with 19/28z CAR T cells compared to those receiving control 4H11/28z CAR T cells, where the levels of Tregs were positively correlated with the survival times in anti-CD19 CAR T cell-treated mice [22].

Recent clinical data emphasizes the importance of the spatiotemporal distribution of CAR T cells as well as the accompanying immune effector cells, especially in the bone marrow and cerebrospinal fluid (CSF), where CAR T cell migration into CSF has been detected as early as 7 days post-infusion [23]. Furthermore, T cell activation facilitates the migration of effector and memory T cells across the blood-brain barrier [24,25]. Recently, Shneider et al. demonstrated the safety and early efficacy of UCB Treg cells in treating the neuro-inflammatory component of amyotrophic lateral sclerosis, where the clinical improvement was correlated with the improvement in neurofilament levels in the plasma and CSF [26]. Therefore, we speculate that the systemic administration of UCB Treg cells might be able to alleviate the CAR T cell-induced neuro-inflammation.

CAR T cell recipients also suffer from hematologic toxicity, including cytopenias, with a reported cumulative 1-year incidence of 58% [27]. Risk factors for cytopenia include severe CRS [28]. In our xenogeneic model, a significant lymphocyte burden was observed in the bone marrow of the CAR T cell recipients, which was resolved by the addition of UCB Treg cells.

Inflammatory cytokines, including MCP-1, SGP130, interferon-gamma, IL-1, Eotaxin, IL-13, IL-10, macrophage inflammatory protein-1 alpha [29,30], as well as IL-6, IL-15, and TGF-β [1], serve as serum biomarkers that have been shown to act as independent predictors of the risk of developing CRS and neurotoxicity, respectively [31]. In our xenogeneic model, several day-14 biomarkers were examined and compared in the different treatment arms, revealing a significant decrease in eotaxin, GM-CSF, IFNα2, IFNγ, IL-13, MCP-1 and -3, and sCD40L in the CAR T + UCB Treg cell recipients when compared to CAR T cells alone.

CCL11/Eotaxin is an important eosinophil-specific chemokine that is associated with the recruitment of eosinophils into sites of inflammation, is well-described in allergic lung diseases, and participates in innate immunity [32]. Eotaxin is produced by IFNγ-stimulated endothelial cells, complement-activated eosinophils, TNF-activated monocytes, and dermal fibroblasts, and its production is also stimulated by IL-4 [32]. More recently, Eotaxins have been shown to play a role in inducing neuronal cytotoxicity effects by promoting the production of reactive oxygen species (ROS) in microglia cells, where elevated plasma levels of CCL11 have been observed in neuroinflammation and neurodegenerative disorders [33]. Therefore, a decrease in Eotaxin levels caused by UCB Treg cells may suggest their role in mitigating CAR T cell-induced neurotoxicity.

Additionally, macrophages have been identified as potential key mediators in CAR T cell-associated CRS [34,35], where GMCSF produced by activated hematopoietic and nonhematopoietic cells and highly expressed in microglia, brain macrophages, and astrocytes may explain the neuro-associated symptoms after CAR T cell therapy [36]. Furthermore, IFN-γ can cause flu-like symptoms and trigger macrophage activation, leading to the secretion of host cytokines such as IL-6, TNF-α, and IL-10, which could further exacerbate CRS [37]. UCB Treg cell-mediated decreases in GMCSF and IFN-γ levels may contribute to reducing CRS. Another important inflammatory mediator, CD154, also known as CD40 ligand, has been identified as a novel binding partner for some members of the integrin family [38], where soluble CD40 ligand (sCD40L) is released from activated platelets and T cells through proteolytic cleavage initiated by the interaction of the ligand with CD40 T-cell surfaces. Increased levels of sCD40L have been reported in many inflammatory disorders [39,40,41], including CRS [19,42]. Again, UCB Treg cells caused a decrease in sCD40L levels and might be able to mitigate CAR T cell-induced CRS.

Although our immune-deficient xenogeneic B-cell lymphoma model does not fully replicate the complexity of the human tumor microenvironment and the inflammatory cascade triggered by CD19 CAR T cells, it does provide proof of concept that UCB Treg cells do not impede the tumor-killing capabilities of CD19 CAR T cells. Our data support further exploration of adjunct therapy with UCB Treg cells administered three days after CD19 CAR T cell infusion in high-risk patients with B-cell lymphoma/leukemia who are prone to developing CRS.

## Figures and Tables

**Figure 1 cells-12-01880-f001:**
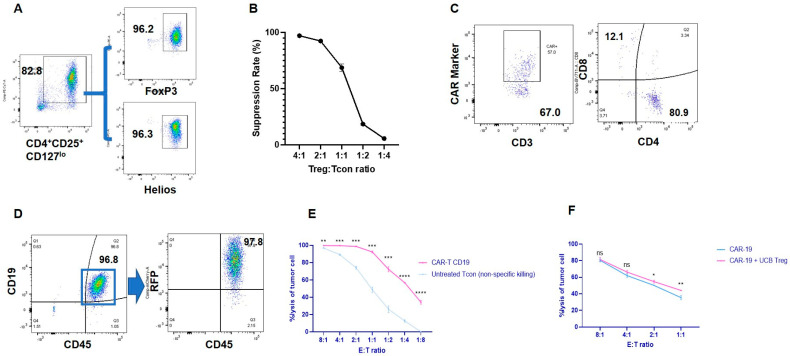
UCB Tregs do not interfere with CAR T cell−induced cytotoxicity. (**A**) UCB Treg phenotype. Freshly harvested UCB Tregs were cultured ex vivo with CD3/28 Dynabeads^TM^ and the continued presence of IL−2 for 14 days. At the time of harvest, phenotype analysis of the cells using Flow cytometry was performed for cell surface expression of CD4^+^CD25^+^CD127^lo^ and intracellular expression of FOXP3^hi^Helios^hi^. (**B**) Cell Suppression Assay. Ex-vivo cultured UCB Tregs were co-cultured with CTV-labeled Tcon (CD4^+^CD25^−^) cells in the presence of CD3/28 beads at different ratios of 4:1, 2:1, 1:1, 1:2, and 1:4 to evaluate their function in Tcon cell proliferation. CTV-labeled Tcons were activated with CD3/CD28 beads at a ratio of 2:1 for CTV-labeled Tcon cells: CD3/CD28 beads and then co-cultured with different ratios of unlabeled UCB-Tregs. Proliferation of CTV-labeled Tcons was assessed by the LSR Fortessa Cell Analyzer after 96 h of culture. Percentage suppression was calculated using the following formula: 100% × (1 − percentage of proliferating CTV-diluting Tcons in the presence of UCB Tregs at a different ratio/percentage of proliferating CTV-diluting Tcons when cultured alone). (**C**) CAR T cell phenotype. Clinical−grade CD19^+^ CAR T cells were analyzed for cell surface expression of CD3, 4, 8, and CAR markers. The two flow cytometry plots are in parallel, and both are based on single−cell gating. This is a one-time quality control test of CAR-T cell samples before animal experiments to determine the baseline expression intensity of CAR, CD4, and CD8. The sample does not contain UCB Treg cells. (**D**) Lymphoma cell phenotype. RFP-labeled CD19^+^ Raji cells were analyzed using flow cytometry for cell surface expression of CD19, CD45, and RFP. The two FACS plots are both based on single-cell gating. (**E**) CART induces tumor cytotoxicity. A flow cytometer-based tumor cytotoxicity assay was performed to analyze the CART-induced tumor cell lysis when compared to Tcon alone as a control. CD19 CART cells demonstrated significantly higher toxicity for CD19-expressing RFP−labeled Raji cells when compared to Tcon at different CAR T: Raji cell ratio of 8:1 (99.6% vs. 96.8%); 4:1 (99.5% vs. 89.1%); 2:1 (98.6% vs. 74.1%); 1:1 (92.2% vs. 48.6%); 1:2 (72.2% vs. 26.1%); 1:4 (56.7% vs. 12.5%); and 1:8 (34.1% vs. 0%). (**F**) UCB Tregs do not interfere with CART-induced tumor cytotoxicity. No impact of the addition of UCB Tregs was observed in the ability of the CD19 CAR T cells to exert cytotoxicity on CD19 Raji cells at different ratios: 8:1 (80.4% vs. 81.5%); 4:1 (62.0% vs. 66.2%); 2:1 (50.1% vs. 54.7%); 1:1 (35.4% vs. 44.1%). (**B**,**E**,**F**) Experiments were performed in triplicate. Error bars represent SEM. Statistical differences compared with PB were quantified by a paired t−test; * *p* < 0.05, ** *p* < 0.01, *** *p* < 0.001, **** *p* < 0.0001. ns = not significant; CTV = CellTrace^TM^ Violet; PB = peripheral blood; SEM = standard error of means.

**Figure 2 cells-12-01880-f002:**
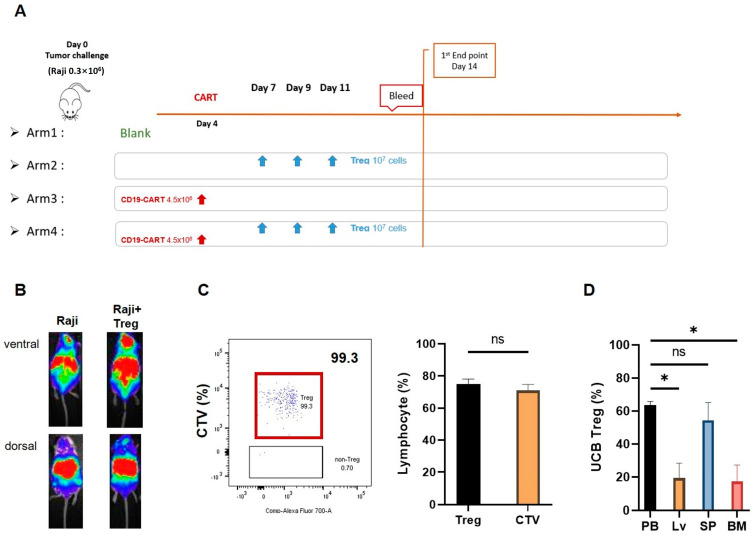
UCB Treg cells show persistence in a xenogeneic lymphoma model. (**A**) Xenogeneic lymphoma model and UCB Treg injection. Female Rag2−γc−mice were injected with 0.3 × 10^6^ RFP−fLuc−Raji cells by tail vein on Day 0. On day 4, after mice displayed engraftment of the tumor mass as per in vivo imaging, a one-time tail vein injection of 4.5 × 10^6^ CD19 CAR-T cells was administered. on day 4, followed by 1 × 10^7^ UCB-Treg cells administration by tail vein injection on days +7, +9, and +11. Peripheral blood and mouse organ tissue samples were harvested from day 14 onward. (**B**) Non-invasive bioluminescence detects tumors in UCB Treg recipients. Single tail vein injection of 0.3 × 10^6^ Raji cells led to the development of disease in NSG mice +/− UCB Treg cells, where tumor engraftment is higher in the areas corresponding to the liver and brain in control mice compared to differential preference for the spleen and PB in the +UCB Treg arm. (**C**) Representative flow cytometric plots show that almost all the circulating non-tumor cell population in Treg recipients was comprised of CTV-labeled cells, and all CTV labeled cells demonstrated the phenotype of Treg cells (CD4^+^CD25^+^), *p* = n.s., student *t*-test. CTV, CellTrace^TM^ Violet; PB, peripheral blood. (**D**) UCB Tregs concentrate in the PB and liver in the xenogeneic lymphoma model. Flow analysis was performed in the harvested organ cell suspension upon euthanasia performed on day 14. The PB and spleen have a higher UCB Treg distribution. Error bars represent SEM (n = 7); statistical differences compared with PB were quantified by one-way ANOVA using GraphPad Prism software (version 9.5.0): * *p* < 0.05.

**Figure 3 cells-12-01880-f003:**
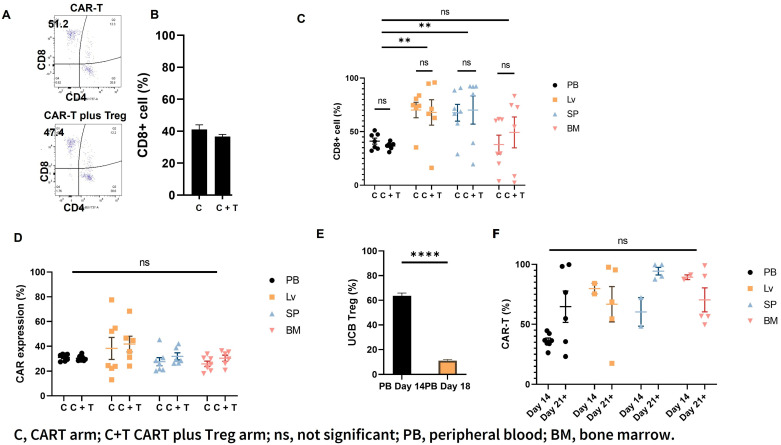
UBC Tregs do not impact CAR T persistence in vivo. (**A**,**B**) CD8^+^ T cells are similar in the PB of CAR T vs. CAR T + UCB Treg recipients. Representative flow cytometric plots of CAR T gating are shown; the circulating CD8^+^ effector T-cell population in the PB of the recipients of CAR T does not differ from CAR T plus UCB Treg cells. (**C**) CD8^+^ T cells are similar in different organs of CAR T vs. CAR T + UCB Treg recipients. No differences were observed in the distribution of the CD8^+^ effector population in liver, spleen, and BM samples of the CAR T (Arm 3; C) vs. CAR T + UCB Treg cell (Arm 4; C+T) recipients. (**D**) CAR expression is similar in different organs of CAR T vs. CAR T + UCB Treg recipients. No difference in CAR expression was observed between the recipients of CAR T (Arm 3; C) vs. CAR T plus UCB Treg cells (Arm 4; C+T) in PB, liver, spleen, or BM samples. (**E**) Circulating UCB Tregs decline over time. The circulating UCB Treg cells remained elevated in the PB circulation until day 14 and dropped to control levels around day 18. Statistical differences compared with PB Day 14 were quantified by the student *t*-test. (**F**) CAR T cells show persistence in CAR T + UCB Treg recipients. The circulating UCB Treg cells did not exert any negative impact on the circulating CAR T cells, as demonstrated by their sustained and continued presence in PB on days +14 and +21. (**C**–**F**) Error bars represent SEM (n = 7); statistical differences compared with PB were quantified by two-way ANOVA using GraphPad Prism software (version 9.5.0): ** *p* < 0.01, **** *p* < 0.0001, ns, not significant. PB, peripheral blood; BM, bone marrow.

**Figure 4 cells-12-01880-f004:**
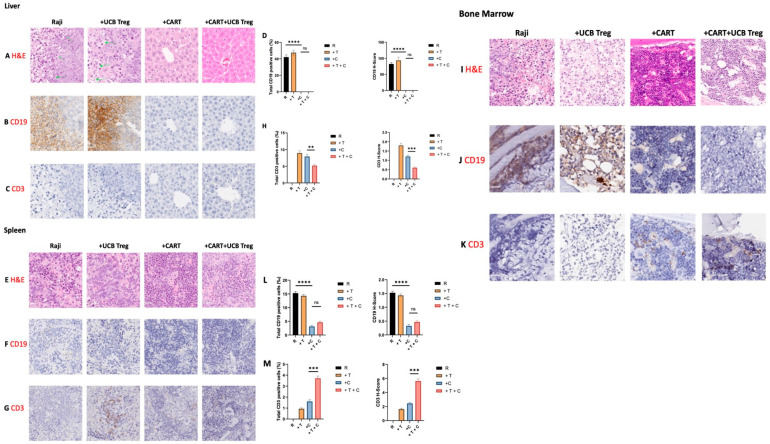
UCB Tregs do not interfere with CAR T cell-mediated tumor clearance. Histopathologic examinations of liver, spleen, and BM tissue show: (**A**–**C**) The liver architecture was disrupted and invaded by the tumor cells in Arm 1: Raji-only arm (row 1; image 1), as well as in Arm 2: Raji + UCB Treg cells (row 1; image 2). Immunohistochemical staining showed the detection of CD19-positive tumor cells in both arms (row 2; images 1 and 2). No evidence of tumor was detected in Arm 3: Raji + CAR T cell recipients or Arm 4: Raji + CAR T + UCB Treg cell recipients with complete preservation of the tissue architecture (rows 1 to 2; images 3 and 4). Specifically, no differences in the CD3^+^ T cell infiltrate were observed in either arm (row 3; images 1–4). (**E**–**G**) Tissue disruption was seen in the spleen of Arm 1: Raji-only mice (row 1; image 1) and Arm 2: Raji + UCB Treg cell recipient (row 1; image 2); however, tumor invasion was not detected by the CD19 IHC stain of the splenic tissue in either arm (row 2; images 1-4). CD3^+^ T cell staining showed some presence in Arm 2: Raji + UCB Treg only and Arm 3: Raji + CAR T recipients, but barely in Arm 4: Raji + CAR T + UCB Treg cell recipients (row 3; images 1–4). (**I**–**K**) Massive tumor infiltration of bone marrow was detected in the Arm 1: Raji only control (row 1; image 1) and Arm 2: Raji + UCB Treg cell recipients (row 1; image 2), whereas no tumor was detected in the Arm 3: Raji + CAR T and Arm 4: Raji + CAR T + UCB Treg cell recipients (row 1; images 3 and 4). A dense lymphocytic infiltration was evident in Arm 3: Raji + CAR T cell recipient (row 1; image 3), whereas the bone marrow architecture was well preserved in Arm 4: Raji + CAR T + UCB Treg cell recipient (row 1; image 4). Immunohistochemical staining showed the detection of CD19-positive tumor cells in Arm 1: Raji-only and Arm 2: Raji + UCB Tregs recipients (row 2; images 1 and 2). No evidence of tumor was detected in Arm 3: Raji + CAR T cell recipients or Arm 4: Raji + CAR T + UCB Treg cell recipients. (**D**,**H**,**L**,**M**) Quantification analysis of the H-score for human CD19 and CD3 positivity. The H-score was defined by the percentage of strongly positive stain × 3 + moderately positive stain × 2 + weakly positive stain × 1. A final value of 0–300 was also calculated at 40× magnification using the software HALO (v3.5-3577.140). A *p* < 0.05 was considered statistically significant. ** *p* < 0.01, *** *p* < 0.001, **** *p* < 0.0001. The statistical differences were quantified by a one-way ANOVA or student’s *t*-test.

**Figure 5 cells-12-01880-f005:**
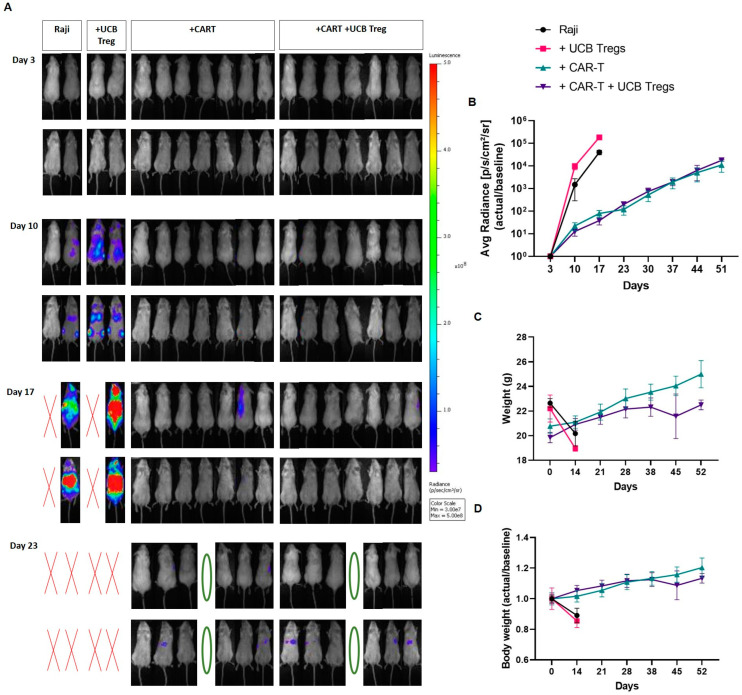
UCB Tregs do not increase tumor burden in CAR T cell recipients with xenogeneic lymphoma. Raji tumor burden in mice was evaluated by Non-invasive bioluminescence imaging (BLI) with the IVIS Lumina X5 Imaging System. Ventral (upper row) and dorsal (lower row) images are shown for each day of capture. (**A**) Non-invasive BLI showed clear evidence of disease progression in Arm 1: Raji controls and Arm 2: Raji + UCB Treg cell recipients, with no evidence of disease in Arm 3: Raji + CAR T and Arm 4: Raji + CAR T + UCB Treg recipients on day 10, where these mice had to be euthanized by day 19. Green circles represent planned euthanasia, and red crosses represent mice found dead. (**B**) Quantitative analysis of the BLI showed no differences measured in photons/sec/cm^2^/sr for Arm 3: Raji + CAR T vs. Arm 4: Raji + CAR T +Tregs up to day 51 follow-up. (**C**) CAR T and CAR T + UCB Tregs maintain body weight. Mouse weight was measured on a weekly basis. Readings in different arms are shown as a linear graph over time. (**D**) Data are shown as fold change in serial days compared with baseline. Error bars represent SEM; statistical differences compared with the control arm were quantified by two-way ANOVA using GraphPad Prism software (version 9.5.0).

**Figure 6 cells-12-01880-f006:**
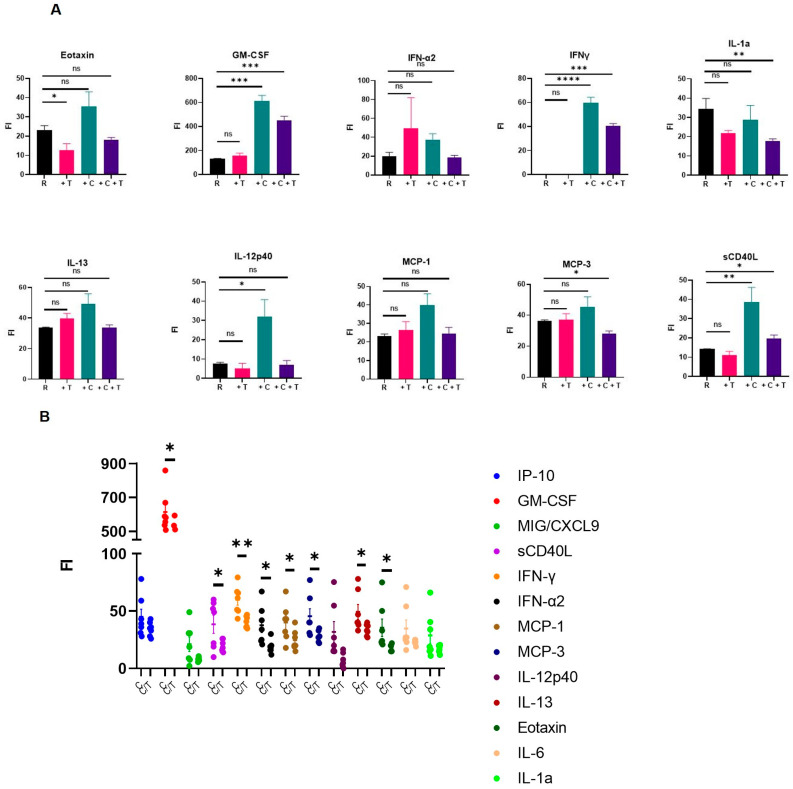
UCB Tregs decrease systemic inflammation in CAR T cell recipients. (**A**). Multiple inflammatory cytokines decreased in UCB Treg recipients: (i) Eotaxin; (ii) GM-CSF; (iii) IFN-α; (iv) IFN-γ; (v) IL-13; (vi) MCP-1; (vii) MCP-3; and (viii) soluble CD40 ligand (sCD40L). (**B**). CAR T vs. CAR T + UCB Tregs shows the anti-inflammatory impact of UCB Tregs. Specifically, when compared to CART recipients, the addition of UCB Treg cells clearly decreased the burden of several inflammatory cytokines. Error bars represent SEM (n = 7); statistical differences compared with the control arm were quantified by one-way ANOVA using GraphPad Prism software: * *p* < 0.05, ** *p* < 0.01, *** *p* < 0.001, **** *p* < 0.0001. PB, peripheral blood; FI, Fluorescent Intensity. The corresponding cytokine concentration of the FI can be calculated with the standard curve. We simply use FI as the unified equivalent of different cytokines.

## Data Availability

All data generated or analyzed during this study are included in this article and Appendix A.

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
