# Peer review of "Adjunct Therapy with T Regulatory Cells Decreases Inflammation and Preserves the Anti-Tumor Activity of CAR T Cells"

_cells, 2023, doi:10.3390/cells12141880_

Round 1

Reviewer 1 Report

Zeng Ke and colleagues report on the effect of T reg umbilical derived for modulation of CAR-T side effects.

This paper is very well designed and written.

I think it deserves to be published!! Well done!

MINOR ISSUES

- "spaciotemporal" wrong spelling

- Discussion is difficult. to read try to simplify and add key messages in bold

Zeng Ke and colleagues report on the effect of T reg umbilical derived for modulation of CAR-T side effects. 

This paper is very well designed and written.

I think it deserves to be published!! Well done!

MINOR ISSUES

- "spaciotemporal" wrong spelling

- Discussion is difficult. to read try to simplify and add key messages in boldZeng Ke and colleagues report on the effect of T reg umbilical derived for modulation of CAR-T side effects. 

This paper is very well designed and written.

I think it deserves to be published!! Well done!

MINOR ISSUES

- "spaciotemporal" wrong spelling

- Discussion is difficult. to read try to simplify and add key messages in bold

Author Response

We thank the reviewer for his encouragement and positive feedback.

We have corrected the spelling mistake of the word "spaciotemporal"

We have heavily revised the manuscript's grammar as well as the discussion section to make is easier and more cohesive. We appreciate the reviewer's insightful comments that helped us to make the manuscript better

Reviewer 2 Report

This is an intriguing study. The hypothesis is that umbilical cord derived  Tregs might have clinical utility in attenuating the (unwanted) inflammation which often accompanies, and can limit the utility of, allogeneic CAR-T cell therapy in cancer.

They test this in an experimental model in vitro with RAJI cells, and in vivo in a RagGammacKO mouse model. In both scenarios the data support their hypothesis. Of course, the data in a clinical model remains to be seen. Moreover, even while waiting for this data it would be of value to see additional studies in a non-xeno model (mouse only), again using a mouse lymphoma/leukemia target and allogeneic murine CAR-T cells with mouse umbilical cord Tregs. Nevertheless the authors are to be credited with a novel idea, and certainly some very promising early studies

Minor changes only...grammar especially!

Author Response

This is an intriguing study. The hypothesis is that umbilical cord derived  Tregs might have clinical utility in attenuating the (unwanted) inflammation which often accompanies, and can limit the utility of, allogeneic CAR-T cell therapy in cancer.

Response: We thank the reviewer for the encouragement and positive feedback. 

They test this in an experimental model in vitro with RAJI cells, and in vivo in a RagGammacKO mouse model. In both scenarios the data support their hypothesis. Of course, the data in a clinical model remains to be seen. Moreover, even while waiting for this data it would be of value to see additional studies in a non-xeno model (mouse only), again using a mouse lymphoma/leukemia target and allogeneic murine CAR-T cells with mouse umbilical cord Tregs.

Response: We thank the reviewer for the insightful comments, especially in regard to examining the role of mouse umbilical cord blood Tregs in mitigating allogeneic murine CAR-T cells mediated CRS in a murine lymphoma model. Unfortunately, at this time, our laboratory does not have access to murine Treg cells and we do not have IACUC approval for murine studies utilizing mouse lymphoma model and mouse CAR T cells. We will apply for these approvals in the future and will try to validate our hypothesis in another publication.   

Nevertheless the authors are to be credited with a novel idea, and certainly some very promising early studies

Response: We thank the reviewer for the encouragement.

Comments on the Quality of English Language

Minor changes only...grammar especially!

Response: We thank the reviewer for the insightful comments. We have heavily revised the grammar and have made the discussion section fluent and easier to read. 

Reviewer 3 Report

Lines 58-59: Rephrase. Tregs are not only a subset of CD4+ T cells, but also include other T cell types such as CD8+ Tregs, γδ Tregs, CD4-CD8- αβ Tregs, HLA-G Tregs, NKTr. I also think that their activity should not be described as “immunosuppressive”, implying a general effect, but as “immunoregulatory”. Moreover, their action is not limited to “maintaining self-tolerance” (this is mainly limited to nTregs), but certain Treg types have the role of dampening the immune response to foreign antigens (e.g. Th3 cells that inhibit an immune response to food antigens and gut bacteria, etc.) and excessive inflammation regardless of its causes.

Lines 62-64: The statement “On the other hand, patients with autoimmune diseases, with defective Tregs are at a higher risk of developing malignancies”, based on ref.7, is not quite accurate. Rather, it depends on the type of autoimmune disease. In patients with certain type-1 autoimmune diseases (characterized by lower numbers and/or dysfunctional Tregs), the incidence of various types of cancers or hematologic malignancies is average or even low, whereas in patients with certain type-2 autoimmune diseases (characterized by normal numbers and functional Tregs), the incidence of hematologic malignancies is higher (see, e.g., for example, Giat et al., Autoimmun. Rev., 2017; Psianou et al., Autoimmun. Rev., 2018; Ghajarzadeh et al., Autoim. Rev., 2020). Rephrase.

Lines 90-92. The statement “The UCB Tregs provided by the two different sources were comparable in phenotype and suppressor function [9]” is confusing because a method to test the function of ex vivo expanded UCB Tregs is described below (lines 115-124). Clarify.

Lines134-140: Explain the cell numbers used in the experiments (for Raji, CAT T cells and UCB Tregs).

In discussion explain how the results obtained with immunodeficient mice can be translated to human studies. Provide a clinical use scenario for the treatment of a B-cell leukemia in the form of a clinical protocol.

Author Response

Comment: Lines 58-59: Rephrase. Tregs are not only a subset of CD4+ T cells, but also include other T cell types such as CD8+ Tregs, γδ Tregs, CD4-CD8- αβ Tregs, HLA-G Tregs, NKTr. I also think that their activity should not be described as “immunosuppressive”, implying a general effect, but as “immunoregulatory”. Moreover, their action is not limited to “maintaining self-tolerance” (this is mainly limited to nTregs), but certain Treg types have the role of dampening the immune response to foreign antigens (e.g. Th3 cells that inhibit an immune response to food antigens and gut bacteria, etc.) and excessive inflammation regardless of its causes.

Response: We agree with the reviewer and have modified the sentence as below:

T regulatory cells (Tregs), an immunoregulatory subset of T cells, play an important role in maintaining self-tolerance, dampening immune response to foreign antigens, resolving excessive inflammation and immunological homeostasis[4, 5].

Comment: Lines 62-64: The statement “On the other hand, patients with autoimmune diseases, with defective Tregs are at a higher risk of developing malignancies”, based on ref.7, is not quite accurate. Rather, it depends on the type of autoimmune disease. In patients with certain type-1 autoimmune diseases (characterized by lower numbers and/or dysfunctional Tregs), the incidence of various types of cancers or hematologic malignancies is average or even low, whereas in patients with certain type-2 autoimmune diseases (characterized by normal numbers and functional Tregs), the incidence of hematologic malignancies is higher (see, e.g., for example, Giat et al., Autoimmun. Rev., 2017; Psianou et al., Autoimmun. Rev., 2018; Ghajarzadeh et al., Autoim. Rev., 2020). Rephrase.

Response. We agree with the reviewer. We have deleted the whole line on autoimmune diseases and cancer since it does not add to the paper and may become confusing to the reader.

Comment: Lines 90-92. The statement “The UCB Tregs provided by the two different sources were comparable in phenotype and suppressor function [9]” is confusing because a method to test the function of ex vivo expanded UCB Tregs is described below (lines 115-124). Clarify.

Response: We have clarified the statement by referencing our previous work where the two sources of cells have been compared and found to be similar as below:

The UCB Tregs provided by the two different sources were comparable in phenotype and suppressor function, as described previously[8].

The method to test the function of ex vivo expanded UCB Tregs has been described to support the function of the UCB Treg cells being injected into the xenogenic model. 

Comment: Lines134-140: Explain the cell numbers used in the experiments (for Raji, CAT T cells and UCB Tregs).

 Response. We have added the explanation for the cell numbers as below:

The dose of Raji cell and CD19 CAR T cell was based on published literature followed by additional titration in our laboratory[17]. The dose of UCB Treg cells was based on our laboratory experience[8].

Comment: In discussion explain how the results obtained with immunodeficient mice can be translated to human studies. Provide a clinical use scenario for the treatment of a B-cell leukemia in the form of a clinical protocol.

Response: We thank the reviewer for bringing up this very important point. We have added the following paragraph at the end of the discussion to address the above item

Although our immune-deficient xenogeneic B cell lymphoma model does not fully replicate the complexity of the human tumor microenvironment and the inflammatory cascade triggered by CD19 CAR T cells, it does provide proof of concept that UCB Treg cells do not impede the tumor-killing capabilities of CD19 CAR T cells. Our data support further exploration of adjunct therapy with UCB Treg cells administered three days after CD19 CAR T cell infusion in high-risk patients with B cell lymphoma/leukemia who are prone to developing CRS.